

# Genetic variation and DNA fingerprinting of durian types in Malaysia using simple sequence repeat (SSR) markers

Ging Yang Siew[1], Wei Lun Ng[1,2,3], Sheau Wei Tan[1], Noorjahan Banu Alitheen[3], Soon Guan Tan[3] and Swee Keong Yeap[1,4]

[1] Institute of Bioscience, Universiti Putra Malaysia, Serdang, Selangor, Malaysia
[2] School of Life Sciences, Sun Yat-sen University, Guangzhou, Guangdong, China
[3] Department of Cell and Molecular Biology, Faculty of Biotechnology and Biomolecular Sciences, Universiti Putra Malaysia, Serdang, Selangor, Malaysia
[4] China-ASEAN College of Marine Sciences, Xiamen University Malaysia, Sepang, Selangor, Malaysia

Corresponding authors
Wei Lun Ng, ng.wl85@gmail.com, ngweilun@mail.sysu.edu.cn
Noorjahan Banu Alitheen, noorjahan@upm.edu.my

## ABSTRACT

Durian (*Durio zibethinus*) is one of the most popular tropical fruits in Asia. To date, 126 durian types have been registered with the Department of Agriculture in Malaysia based on phenotypic characteristics. Classification based on morphology is convenient, easy, and fast but it suffers from phenotypic plasticity as a direct result of environmental factors and age. To overcome the limitation of morphological classification, there is a need to carry out genetic characterization of the various durian types. Such data is important for the evaluation and management of durian genetic resources in producing countries. In this study, simple sequence repeat (SSR) markers were used to study the genetic variation in 27 durian types from the germplasm collection of Universiti Putra Malaysia. Based on DNA sequences deposited in Genbank, seven pairs of primers were successfully designed to amplify SSR regions in the durian DNA samples. High levels of variation among the 27 durian types were observed (expected heterozygosity, $H_E = 0.35$). The DNA fingerprinting power of SSR markers revealed by the combined probability of identity (PI) of all loci was $2.3 \times 10^{-3}$. Unique DNA fingerprints were generated for 21 out of 27 durian types using five polymorphic SSR markers (the other two SSR markers were monomorphic). We further tested the utility of these markers by evaluating the clonal status of shared durian types from different germplasm collection sites, and found that some were not clones. The findings in this preliminary study not only shows the feasibility of using SSR markers for DNA fingerprinting of durian types, but also challenges the current classification of durian types, e.g., on whether the different types should be called "clones", "varieties", or "cultivars". Such matters have a direct impact on the regulation and management of durian genetic resources in the region.

# INTRODUCTION

Durian (*Durio zibethinus*) belongs to the family Malvaceae and is distinctively characterized by its large fruit size, unique odor when ripe, large seeds covered with fleshy or leathery

arils, as well as a thorn-covered husk (*Integrated Taxonomic Information System, 2017*; *Nyffeler & Baum, 2001*). It is diploid with a chromosome number of $n = 28$ (*Brown, 1997*). A recent study that reported the draft genome of durian estimated its genome size to be approximately 738 Mb (*Teh et al., 2017*). Owing to its self-incompatibility, durian is mainly outcrossing, with fruit bats serving as its main pollinator in nature (*Bumrungsri et al., 2009*). In the genus *Durio*, a total of 34 species are known (*The Plant List, 2013*), and at least nine of them produce edible fruits (*Idris, 2011*). Of the nine species, *D. zibethinus* is the most common and is often cultivated in home gardens or orchards.

Popularly known as the "King of Fruits", durian is one of the most popular tropical fruits in Asia. Believed to have originated from Borneo (*Morton, 1987*; *Tarmizi & Abidin, 1991*), durian is widely cultivated in countries located near the equator such as Malaysia, Indonesia, Thailand, Myanmar, the Philippines, Sri Lanka, India, Australia, and Papua New Guinea (*Tarmizi & Abidin, 1991*), and is found wild or semi-wild in many countries around South and Southeast Asia (*Morton, 1987*). Two of the largest exporters of durian in the world are Malaysia and Thailand (*Siriphanich, 2011*). Durian from Malaysia, for example, is exported to many countries including Singapore, Indonesia, Hong Kong, and China, which are the top four importers in 2015. The export value to these countries alone in 2015 totaled approximately USD 14.8 million (Department of Agriculture Malaysia, pers. comm., 2016).

Durian is classified into different "clones" or "varieties" (or "cultivars"), based on phenotypic characters of the fruit. While cultivated durian is mostly asexually propagated (*Brown, 1997*), so far no study has evaluated the clonality of cultivated durian. For consistency, and to remain neutral at this stage, we shall use the term "durian type" throughout this paper. In Malaysia, 126 durian types have been registered with the Department of Agriculture Malaysia, as of September 2017 (*Department of Agriculture Malaysia, 2017*), based on fruit shape, thorn size, aroma of the fruit, and seed shape (*Department of Agriculture Malaysia, 2010*) . Morphological characters are easy to observe, fast, and cheap but they suffer from phenotypic plasticity as a direct result of environmental factors (e.g., climate, nutrient and moisture content, and soil type) and age, which may contribute to morphological variation (*Chambel et al., 2005*). To overcome the limitation of phenotypic plasticity, there is a need to carry out genetic characterization on the registered durian types.

Recently, there have been studies on the genetic variation of durian types from important durian producing countries using DNA markers such as inter-simple sequence repeat (ISSR) (*Siew et al., 2018*; *Vanijajiva, 2012*) and random amplified polymorphic DNA (RAPD) (*Vanijajiva, 2011*; *Ruwaida, Supriyadi & Parjanto, 2009*) markers. While the ease of application of these markers makes them attractive choices for studies on overall genetic variation and population genetic structure (*Ng & Tan, 2015*), the dominant nature of these markers do not work well with applications such as DNA fingerprinting (*Kirst et al., 2005*). Moreover, the data generated from dominant genetic markers are not as informative as co-dominant markers and some are known to suffer from poor reproducibility (*Semagn, Bjørnstad & Ndjiondjop, 2006*), throwing into question the feasibility and reliability of using such markers for downstream applications. Simple sequence repeat (SSR) markers,

on the other hand, are codominant, multi-allelic, and highly reproducible. They are one of the most powerful markers for plant variety identification and have been successfully applied to study genetic variation in a wide range of cultivated plant species such as oil camellia (*Camellia oleifera*; *Chen et al., 2016*), rice (*Oryza sativa*; *Sarao et al., 2009*), and jute (*Corchorus* spp.; *Zhang et al., 2015*). The availability of markers that generate highly accurate and reproducible results is important for the evaluation and subsequent management of genetic resources.

To our knowledge, few studies have used SSR markers to study the genetic variation in durian (e.g., *Sales, 2015*; *Santoso et al., 2017*). In this study, SSR markers were designed from publicly available DNA sequences containing SSR regions, and used to study the genetic variation among major durian types found in Malaysia. We also evaluated the feasibility of using these markers to genetically fingerprint the various durian types. Finally, we determined the clonality of several durian types sampled from different collection sites, and discuss the implications of our findings toward the regulation and management of durian genetic resources in the region.

## MATERIALS AND METHODS

### Sampling and DNA extraction

Leaves from a total of 45 durian trees were collected across five durian orchards (that also serve as germplasm collection sites) Ekspo Plot A (BEA), Putra Mart (PM), Ladang Puchong (LP), and Ladang 5 (5L) (Table 1). These durian trees have been pre-identified and pre-labeled for the types of durian fruit that they produce. The experimental material consist of 27 samples that represent different durian types, and 18 samples that represent replicates of some of the durian types (i.e., D2, D7, D8, D24, D99, D159, D168, D188, and D197) from different orchards. Many of the sampled durian types in this study are popular commercial types (e.g., D24, D160, D168, and D197; Department of Agriculture Malaysia, pers. comm., 2017), and most have not been studied for genetic diversity using SSR markers.

For DNA extraction, 100 mg of fresh leaf material was ground to powder in liquid nitrogen. Genomic DNA was extracted from the ground leaf material using the cetyl trimethylammonium bromide (CTAB) extraction method as described by *Doyle & Doyle (1990)*. The crude DNA extract was further purified using the GF-1 Plant DNA Extraction Kit (Vivantis Technologies Sdn. Bhd., Malaysia) before further analyses. The purified DNA was quantified using a Nanodrop spectrophotometer (Beckman Coulter, Brea, CA, USA).

### Selection of SSR primers and detection of PCR products

Eight pairs of SSR primers were designed from seven DNA sequences containing SSR regions that were deposited in Genbank, using Primer-BLAST (*Ye et al., 2012*). Detailed primer sequences and their sources are listed in Table 2. PCR was conducted in 20 uL reaction mixtures containing $1\times$ NEXpro$^{TM}$ e PCR Master Mix (Genes Laboratories, Bokjeong-dong, South Korea), 0.2 $\mu$M each of the forward and reverse primers, and approximately 20 ng of genomic DNA. The designed primers were initially tested on two durian DNA samples using two types of PCR protocols on a thermocycler. The first PCR

**Table 1  Details of durian samples used in this study.**

| No. | Type | Common name | No. of samples (sampling location[a]) | Place of origin |
|---|---|---|---|---|
| 1 | D2 | Dato' Nina | 4 (PM, LP, BE, BEA) | Melaka |
| 2 | D7 | N/A | 4 (LP, 5L, BE, BEA) | Selangor |
| 3 | D8 | N/A | 1 (LP) | Kuala Lumpur |
| 4 | D10 | Durian Hijau | 1 (PM) | Selangor |
| 5 | D16 | N/A | 1 (BEA) | N/A |
| 6 | D24 | N/A | 5 (PM, LP, 5L, BE, BEA) | Perak |
| 7 | D84 | N/A | 1 (5L) | Perak |
| 8 | D88 | Bangkok 8 | 1 (5L) | Selangor |
| 9 | D96 | Bangkok A | 3 (PM, LP, 5L) | Selangor |
| 10 | D99 | Kop Kecil | 3 (PM, LP, 5L) | Thailand |
| 11 | D125 | Kop Jantung | 1 (5L) | Kedah |
| 12 | D145 | Tuan Mek Hijau/Beserah | 1 (LP) | Pahang |
| 13 | D148 | Paduka | 1 (LP) | Perak |
| 14 | D158 | Kan Yau/Tangkai Panjang | 1 (LP) | Kedah |
| 15 | D159 | Mon Thong/Bantal Mas | 1 (LP) | Kedah |
| 16 | D160 | Buluh Bawah | 1 (LP) | Selangor |
| 17 | D162 | Tawa | 1 (LP) | Selangor |
| 18 | D168 | Durian Mas Hjh. Hasmah | 3 (PM, LP, 5L) | Johor |
| 19 | D169 | Tok LiTok | 1 (LP) | Kelantan |
| 20 | D172 | Durian Botak | 1 (LP) | Johor |
| 21 | D175 | Udang Merah | 1 (LP) | Pulau Pinang |
| 22 | D188 | MDUR 78 | 2 (LP, BE) | Terengganu |
| 23 | D189 | MDUR 79 | 1 (LP) | Terengganu |
| 24 | D190 | MDUR 88 | 1 (PM) | Terengganu |
| 25 | D197 | Raja Kunyit/Musang King | 2 (PM, LP) | Kelantan |
| 26 | Durian Gergasi (DG) | N/A | 1 (LP) | N/A |
| 27 | Durian Siam (DS) | N/A | 1 (BEA) | N/A |

**Notes.**

Information of the common name and the place of origin are based on the records of Department of Agriculture (*Department of Agriculture Malaysia, 2017*); N/A, Not available.

[a]PM, Putra Mart; LP, Ladang Puchong; BE, Bukit Ekspo; BEA, Bukit Ekspo Plot A; 5L, Ladang 5.

profile consists of an initial denaturation of 3 min at 95 °C, followed by 30 cycles of 30 s at 95 °C, 30 s at 55 °C or 60 °C, and 2 min at 72 °C followed by an extension step at 72 °C for 7 min; and the second PCR used a touch-down protocol that started with an initial denaturation of 3 min at 95 °C, then 10 cycles of 30 s at 95 °C, 30 s at 60 °C (−1 °C/cycle), and 1 min at 72 °C, followed by 25 cycles of 30 s at 95 °C, 30 s at 50 °C, and 1 min at 72 °C, with a final extension step at 72 °C for 7 min. Resultant PCR amplicons for each marker were Sanger-sequenced on an ABI 3730 sequencer, through services provided by First Base Laboratories Sdn Bhd. (Selangor, Malaysia), in order to verify that the amplicons were the targeted regions that contained SSR sequences. Markers that worked well and the corresponding PCR conditions were subsequently used to genotype all durian samples. PCR amplicons were analyzed through electrophoresis on 8% (w/v) polyacrylamide gels, stained with ethidium bromide and viewed under UV illumination. The DNA fragment sizes were

**Table 2 SSR primers used in this study.**

| Locus | Primer name | Primer sequence (5′ → 3′) | Accession number of source sequence on Genbank | Successful amplification of intended fragment? |
|---|---|---|---|---|
| DZ01 | DZ01_F2 | AATTCCACATGACAGACAGG | AB292171 | Yes |
| | DZ01_R | TCATGGATGTTGTATGGCAG | | |
| DZ02 | DZ02_F | ACCTTCTCCCCATTTCACC | AB292166 | Yes |
| | DZ02_R | TGTTGAAGTCATACGTTTAGCC | | |
| DZ03 | DZ03_F | CTCTAAAAAGAATGGGGATATTG | AB292168 | Yes |
| | DZ03_R | ATTCTGGAACAAAAGTTACAAAC | | |
| DZ04 | DZ04_F2 | TGCATGTTTTGAAAAGTACC | AB292170 | Yes |
| | DZ04_R2 | ATGGGGAAAAGAAAGTGAAG | | |
| DZ05 | DZ05_F2 | ACACATACACAACTCACCTC | AB292169 | Yes |
| | DZ05_R | ATGCCCGATGAAATTGTAAC | | |
| DZ06 | DZ06_F | ATGGGATTTGGATGATGGGTTG | AB292165 | No |
| | DZ06_R | CGACTCACTATAGGGCGAATTG | | |
| | DZ06_F2 | AGGTTGAATTGAACTGGGTTTTG | | |
| | DZ06_R2 | GCGGGAATTCGATTGATGAG | | |
| DZ07 | DZ07_F | ACACACCATCTTCCCTTTG | AB292167 | Yes |
| | DZ07_R | TGCACATGTTGTTTGTATATATG | | |
| DZ08 | DZ08_F | ACATATATACAAACAACATGTGC | AB292167 | Yes |
| | DZ08_R2 | GTCCAATGATGGAAAAACTC | | |

estimated by comparison of sample banding patterns with a 50 bp DNA ladder (New England Biolabs Inc., Ipswich, MA, USA) loaded in the same gel. PCR and polyacrylamide gel electrophoresis were repeated to ensure consistency of the results.

## Data analysis
### Genetic variability and fingerprinting
The estimation of genetic variability and fingerprinting power was conducted on the 27 durian samples representing different durian types. The estimated DNA fragment sizes of each sample at each locus were manually recorded. GenAlEx 6.502 (*Peakall & Smouse, 2012*) was used to estimate basic genetic parameters, such as the total number of alleles, number of alleles per locus, allele frequency, as well as the expected ($H_E$) and observed ($H_O$) heterozygosities.

The probability of identity (PI) of each marker and of the combination of all loci were calculated using GenAlEx 6.502 (*Peakall & Smouse, 2012*) to assess the fingerprinting power of the SSR markers. The DNA fragments obtained from seven pairs of SSR primers were used for DNA fingerprinting. The amplified fragments of SSRs were encoded 0 for absence of a band and 1 for presence of a band for an allele using GenAlEx 6.502 (*Peakall & Smouse, 2012*).

The same markers were also used to genotype 18 additional samples representing replicates of some of the durian types (i.e., D2, D7, D8, D24, D99, D159, D168, D188, and

**Table 3  Genetic variability and fingerprinting power of the seven SSR markers used in this study.**

| Locus | Number of alleles | Allele | Allele frequency | $H_E$ | $H_O$ | PI |
|---|---|---|---|---|---|---|
| DZ01 | 4 | 210 | 0.074 | 0.615 | 0.519 | 0.2 |
| | | 226 | 0.222 | | | |
| | | 250 | 0.148 | | | |
| | | 260 | 0.556 | | | |
| DZ02 | 5 | 320 | 0.019 | 0.501 | 0.259 | 0.28 |
| | | 340 | 0.093 | | | |
| | | 350 | 0.685 | | | |
| | | 360 | 0.111 | | | |
| | | 376 | 0.093 | | | |
| DZ03 | 3 | 126 | 0.167 | 0.575 | 0.222 | 0.25 |
| | | 140 | 0.574 | | | |
| | | 150 | 0.259 | | | |
| DZ04 | 3 | 200 | 0.37 | 0.621 | 0.667 | 0.22 |
| | | 210 | 0.167 | | | |
| | | 226 | 0.463 | | | |
| DZ05 | 1 | 200 | 1 | 0 | 0 | 1 |
| DZ07 | 1 | 440 | 1 | 0 | 0 | 1 |
| DZ08 | 2 | 140 | 0.926 | 0.137 | 0 | 0.75 |
| | | 160 | 0.074 | | | |
| Mean (excluding monomorphic loci) | 2.714 | – | – | 0.35 (0.49) | 0.238 (0.42) | – |
| Combined | – | – | – | – | – | $2.3 \times 10^{-3}$ |

D197) obtained from different orchards. DNA fingerprints were generated as above and compared among samples of the same durian type.

## RESULTS

### SSR data analysis

Of the eight SSR primer pairs designed, seven primer pairs successfully amplified clear and reproducible bands in all 27 durian types. Five loci were polymorphic and two loci were monomorphic. A total of 19 alleles were scored across seven SSR loci, ranging from one to five alleles per locus with an average of 2.714 alleles per locus. The allele frequency of each allele at each locus ranged from 0.074 to 1. The $H_O$ ranged from 0 to 0.667 with a mean $H_O$ of 0.238, while the $H_E$ ranged from 0 to 0.621 with a mean $H_E$ of 0.35. The $H_E$ was generally higher than $H_O$ at all loci except DZ04. Excluding monomorphic loci, the mean $H_O$ was 0.42, while the mean $H_E$ was 0.49. Detailed results are presented in Table 3.

### DNA fingerprinting power

A total of 17 polymorphic bands were obtained from the seven SSR loci. The PI of each locus and the PI estimated using all loci (hereinafter, 'total PI') were calculated to assess the fingerprinting power of the markers (Table 3). For each locus, the PI value ranged from 0.2 to 1. Assuming that there was no linkage disequilibrium and all loci segregated independently, the chance of finding samples with identical fingerprints is equal to the

total PI for all loci, which is $2.3 \times 10^{-3}$. When only one locus was involved, zero to four (0–14.81%) durians types had distinct fingerprint profiles; when two loci were included, zero to 13 (0–48.15%) durian types had distinct fingerprint profiles; when three loci were included, zero to 21 (0–77.78%) durian types were identified; when four loci were included, two to 21 (7.41–77.78%) durian types were identified; when five loci were included, nine to 21 (33.33–77.78%) durian types were identified; when six loci were included, 16 to 21 (59.26–77.78%) durian types were identified; when all seven loci were included, 21 (77.78%) durian types were identified. The remaining six (22.22%) durian types did not have unique fingerprints: D2 shared the same fingerprint with D10, D7 shared the same fingerprint as D188, and D168 shared the same fingerprint as D197. The results implied that seven SSR markers have successfully fingerprinted 21 out of 27 durian types tested in this study. Detailed results are presented in Tables 4–6.

**Fingerprinting of durian types across orchards**
A total of nine durian types (i.e., D2, D24, D99, D168, D197, D159, D188, D7, and D8) across five orchards in UPM were investigated. Six types (i.e., D2, D99, D197, D159, D188, and D7) were found to contain samples with different fingerprint profiles, with alleles differing at one or more loci. Only three types (i.e., D24, D168, and D8) were found to have the same fingerprint profiles across orchards.

Four samples of D2 from orchards PM, LP, BE, and BEA had different alleles at the locus DZ02. Three samples of D99 from orchards PM, LP, and 5L had different alleles at three loci, i.e., loci DZ01, DZ02, and DZ04. Two samples of D197 from orchards PM and LP had different alleles at locus DZ04. Two samples of D159 from orchards LP and 5L had different alleles at three loci, i.e., loci DZ01, DZ03, DZ04, and DZ08. Two samples of D188 from LP and BE were different at most of the loci, i.e., loci DZ01, DZ02, DZ03, DZ04 and DZ08. Lastly, four samples of D7 from orchards LP, 5L, BE, and BEA had different alleles at two loci, i.e., loci DZ01 and DZ03. The results are summarized in Table 7. This showed that many durian types had different genotypes across orchards.

## DISCUSSION

As far as we are aware, this is one of few studies that have used SSR markers to evaluate genetic variation in durian. A study by *Santoso et al. (2017)* reported the development of SSR markers for the study of genetic variation in durian. However, none of the 11 markers reported contained perfect repeat motifs. Homoplasy has been found to be common with imperfect repeats, i.e., compound and/or interrupted repeats (*Adams, Brown & Hamilton, 2004*), which biases the estimation of genetic variation (*Selkoe & Toonen, 2006*) and renders those markers unsuitable for DNA fingerprinting.

*Sales (2015)* reported the evaluation of 127 sets of SSR primers on 187 durian types. In the current study, we synthesized and pretested the 29 primer pairs recommended in *Sales (2015)* on our durian DNA samples, but none of the primers amplified specific fragments containing SSRs. The primers used in the study were initially developed for cotton (*Gossypium* spp.), explaining the poor transferability of the primers to durian. SSR markers

**Table 4  Number of durian types differentiated based on different marker combinations.**

| Marker combinations | No. durian types differentiated |
|---|---|
| One marker | |
| DZ01 | 0 |
| DZ02 | 4 |
| DZ03 | 2 |
| DZ04 | 0 |
| DZ05 | 0 |
| DZ07 | 0 |
| DZ08 | 0 |
| Two markers | |
| DZ01, DZ02 | 13 |
| DZ01, DZ03 | 10 |
| DZ01, DZ04 | 9 |
| DZ01, DZ05 | 0 |
| DZ01, DZ07 | 0 |
| DZ01, DZ08 | 2 |
| DZ02, DZ03 | 12 |
| DZ02, DZ04 | 11 |
| DZ02, DZ05 | 4 |
| DZ02, DZ07 | 4 |
| DZ02, DZ08 | 6 |
| DZ03, DZ04 | 7 |
| DZ03, DZ05 | 2 |
| DZ03, DZ07 | 2 |
| DZ03, DZ08 | 2 |
| DZ04, DZ05 | 0 |
| DZ04, DZ07 | 0 |
| DZ04, DZ08 | 2 |
| DZ05, DZ07 | 0 |
| DZ05, DZ08 | 0 |
| DZ07, DZ08 | 0 |
| Three markers | |
| DZ01, DZ02, DZ03 | 19 |
| DZ01, DZ02, DZ04 | 17 |
| DZ01, DZ02, DZ05 | 13 |
| DZ01, DZ02, DZ07 | 13 |
| DZ01, DZ02, DZ08 | 13 |
| DZ01, DZ03, DZ04 | 21 |
| DZ01, DZ03, DZ05 | 10 |
| DZ01, DZ03, DZ07 | 10 |
| DZ01, DZ03, DZ08 | 12 |
| DZ01, DZ04, DZ05 | 9 |
| DZ01, DZ04, DZ07 | 9 |

**Table 4** (*continued*)

| Marker combinations | No. durian types differentiated |
|---|---|
| DZ01, DZ04, DZ08 | 11 |
| DZ01, DZ05, DZ07 | 0 |
| DZ01, DZ05, DZ08 | 2 |
| DZ01, DZ07, DZ08 | 2 |
| DZ02, DZ03, DZ04 | 16 |
| DZ02, DZ03, DZ05 | 12 |
| DZ02, DZ03, DZ07 | 12 |
| DZ02, DZ03, DZ08 | 14 |
| DZ02, DZ04, DZ05 | 11 |
| DZ02, DZ04, DZ07 | 11 |
| DZ02, DZ04, DZ08 | 11 |
| DZ02, DZ05, DZ07 | 4 |
| DZ02, DZ05, DZ08 | 14 |
| DZ03, DZ04, DZ05 | 7 |
| DZ03, DZ04, DZ07 | 7 |
| DZ03, DZ04, DZ08 | 9 |
| DZ04, DZ05, DZ07 | 0 |
| DZ04, DZ07, DZ08 | 2 |
| DZ05, DZ07, DZ08 | 0 |
| Four markers | |
| DZ01, DZ02, DZ03, DZ04 | 21 |
| DZ01, DZ02, DZ03, DZ05 | 19 |
| DZ01, DZ02, DZ03, DZ07 | 19 |
| DZ01, DZ02, DZ03, DZ08 | 19 |
| DZ01, DZ02, DZ04, DZ05 | 17 |
| DZ01, DZ02, DZ04, DZ07 | 17 |
| DZ01, DZ02, DZ04, DZ08 | 17 |
| DZ01, DZ02, DZ05, DZ07 | 13 |
| DZ01, DZ02, DZ05, DZ08 | 13 |
| DZ01, DZ02, DZ07. DZ08 | 13 |
| DZ01, DZ03, DZ04, DZ05 | 21 |
| DZ01, DZ03, DZ04, DZ07 | 21 |
| DZ01, DZ03, DZ04, DZ08 | 21 |
| DZ01, DZ03, DZ05, DZ07 | 21 |
| DZ01, DZ03, DZ05, DZ08 | 21 |
| DZ01, DZ03, DZ07, DZ08 | 21 |
| DZ01, DZ04, DZ05, DZ07 | 9 |
| DZ01, DZ04, DZ05, DZ08 | 11 |
| DZ01, DZ05, DZ07, DZ08 | 3 |
| DZ02, DZ03, DZ04, DZ05 | 16 |
| DZ02, DZ03, DZ04, DZ07 | 16 |
| DZ02, DZ03, DZ04, DZ08 | 16 |
| DZ02, DZ03, DZ05, DZ07 | 12 |
| DZ02, DZ03, DZ05, DZ08 | 14 |
| DZ02, DZ03, DZ07, DZ08 | 14 |

**Table 4** (*continued*)

| Marker combinations | No. durian types differentiated |
|---|---|
| DZ02, DZ04, DZ05, DZ07 | 11 |
| DZ02, DZ04, DZ05, DZ08 | 11 |
| DZ02, DZ04, DZ07, DZ08 | 11 |
| DZ03, DZ04, DZ05, DZ07 | 7 |
| DZ03, DZ04, DZ05, DZ08 | 11 |
| DZ04, DZ05, DZ07, DZ08 | 2 |
| Five markers | |
| DZ01, DZ02, DZ03, DZ04, DZ05 | 21 |
| DZ01, DZ02, DZ03, DZ04, DZ07 | 21 |
| DZ01, DZ02, DZ03, DZ04, DZ08 | 21 |
| DZ01, DZ02, DZ03, DZ05, DZ07 | 19 |
| DZ01, DZ02, DZ03, DZ05, DZ08 | 19 |
| DZ01, DZ02, DZ03, DZ07, DZ08 | 19 |
| DZ01, DZ02, DZ04, DZ05, DZ07 | 17 |
| DZ01, DZ02, DZ04, DZ05, DZ08 | 17 |
| DZ01, DZ03, DZ04, DZ05, DZ07 | 21 |
| DZ01, DZ03, DZ04, DZ05, DZ08 | 21 |
| DZ01, DZ03, DZ04, DZ07, DZ08 | 21 |
| DZ01, DZ03, DZ05, DZ07, DZ08 | 12 |
| DZ01, DZ04, DZ05, DZ07, DZ08 | 11 |
| DZ02, DZ03, DZ04, DZ05, DZ07 | 16 |
| DZ02, DZ03, DZ04, DZ05, DZ08 | 16 |
| DZ02, DZ03, DZ04, DZ07, DZ08 | 16 |
| DZ02, DZ03, DZ05, DZ07, DZ08 | 14 |
| DZ02, DZ04, DZ05, DZ07, DZ08 | 11 |
| DZ03, DZ04, DZ05, DZ07, DZ08 | 9 |
| Six markers | |
| DZ01, DZ02, DZ03, DZ04, DZ05, DZ07 | 21 |
| DZ01, DZ02, DZ03, DZ04, DZ05, DZ08 | 21 |
| DZ01, DZ02, DZ03, DZ05, DZ07, DZ08 | 19 |
| DZ01, DZ02, DZ04, DZ05, DZ07, DZ08 | 17 |
| DZ01, DZ03, DZ04, DZ05, DZ07, DZ08 | 21 |
| DZ02, DZ03, DZ04, DZ05, DZ07, DZ08 | 16 |
| Seven markers | |
| DZ01, DZ02, DZ03, DZ04, DZ05, DZ07, DZ08 | 21 |

have been known to be transferable across species within a genus (*Gonçalves-Vidigal & Rubiano, 2011*; *Hodel et al., 2016*; *Selkoe & Toonen, 2006*), but cases of transferability across higher taxonomic levels are rare.

## Genetic variation

$H_E$ is one of the most important and commonly used estimators of genetic diversity when using codominant markers such as SSR markers (*Bashalkhanov, Pandey & Rajora, 2009*; *Nybom, 2004*). A high level of genetic diversity among durian types was observed in this

**Table 5  DNA fingerprint profiles of 27 durian types in fragment sizes.**

| Durian type | DNA fingerprint profile | Shared/unique |
|---|---|---|
| D2 | 260260350350140140200210200200440440140140 | Shared (with D10) |
| D7 | 210260350350150150200226200200440440140140 | Shared (with D188) |
| D8 | 226226350350150150200226200200440440140140 | Unique |
| D10 | 260260350350140140200210200200440440140140 | Shared (with D2) |
| D16 | 260260350350140140200200200200440440140140 | Unique |
| D24 | 250260320360140140210226200200440440140140 | Unique |
| D84 | 260260350376150150226226200200440440160160 | Unique |
| D88 | 226260350350126126200226200200440440140140 | Unique |
| D96 | 260260350350150150200210200200440440140140 | Unique |
| D99 | 260260350350140140226226200200440440140140 | Unique |
| D125 | 226260350350140140200226200200440440140140 | Unique |
| D145 | 226260350376126126200200200200440440140140 | Unique |
| D148 | 226250350360140150200200200200440440140140 | Unique |
| D158 | 260260340360126140200226200200440440140140 | Unique |
| D159 | 260260376376140140210226200200440440140140 | Unique |
| D160 | 250260350376140140200226200200440440140140 | Unique |
| D162 | 250250350350140140200200200200440440140140 | Unique |
| D168 | 226260350350140140210226200200440440140140 | Shared (with D197) |
| D169 | 226226360360140140200226200200440440140140 | Unique |
| D172 | 226250340340126140210226200200440440160160 | Unique |
| D175 | 250250340340126140226226200200440440140140 | Unique |
| D188 | 210260350350150150200226200200440440140140 | Shared (with D7) |
| D189 | 210260350360150150226226200200440440140140 | Unique |
| D190 | 210260350350140140226226200200440440140140 | Unique |
| D197 | 226260350350140140210226200200440440140140 | Shared (with D168) |
| DG | 260260350350150126150210226200200440440140140 | Unique |
| DS | 226260350350150126140200226200200440440140140 | Unique |

**Notes.**
DG, Durian Gergasi; DS, Durian Siam.

study, partly due to the outbreeding nature of the species (*Asrul & Sarip, 2009*). Such a level of genetic diversity was comparable to that of some cultivated fruit plants such as coconut (*Cocos nucifera*, mean $H_E = 0.377$; *Liu et al., 2011*), but lower than that found in other wild fruit species such as wild banana (*Musa balbisiana*, mean $H_E = 0.817$; *Ravishankar et al., 2013*). This is reasonable as only certain durian types are preferentially grown. The genetic diversity estimates could also be affected by sample sizes and numbers of loci used in different studies and sample size is one of the most important factors affecting genetic diversity within population (*Bashalkhanov, Pandey & Rajora, 2009*) as it directly affects the number of scored alleles which is used to measure $H_E$. Furthermore, the loci chosen for a study might have a negative impact on the mean $H_E$ if the loci were monomorphic (*Nybom, 2004*). This could be clearly observed in this study as there were two monomorphic loci. If the two monomorphic loci were excluded, the mean $H_E$ in this study increased from 0.35 to 0.49 in this study.

**Table 6   DNA fingerprint profiles of 27 durian types in binary.**

| Durian type | DNA fingerprint profile | Unique/Shared |
|---|---|---|
| D2 | 0001001000101101110 | Shared (with D10) |
| D7 | 1001001000011011110 | Shared (with D188) |
| D8 | 0100001000011011110 | Unique |
| D10 | 0001001000101101110 | Shared (with D2) |
| D16 | 0001001000101001110 | Unique |
| D24 | 0011100100100111110 | Unique |
| D84 | 0011001010010011101 | Unique |
| D88 | 0101001001001011110 | Unique |
| D96 | 0001001000011101110 | Unique |
| D99 | 0001001000100011110 | Unique |
| D125 | 0101001000101011110 | Unique |
| D145 | 0101001011001001110 | Unique |
| D148 | 0110001100111001110 | Unique |
| D158 | 0001010101101011110 | Unique |
| D159 | 0001000010100111110 | Unique |
| D160 | 0011001010101011110 | Unique |
| D162 | 0010001000101001110 | Unique |
| D168 | 0101001000100111110 | Shared (with D197) |
| D169 | 0100000100101011110 | Unique |
| D172 | 0110010001100111101 | Unique |
| D175 | 0010010001100011110 | Unique |
| D188 | 1001001000011011110 | Shared (with D7) |
| D189 | 1001001100010011110 | Unique |
| D190 | 1001001000100011110 | Unique |
| D197 | 0101001000100111110 | Shared (with D168) |
| DG | 0001001001010111110 | Unique |
| DS | 0101001001101011110 | Unique |

**Notes.**
DG, Durian Gergasi; DS, Durian Siam.

## DNA fingerprinting using SSR markers

DNA fingerprinting power is calculated via the total PI of all loci. The lower the total PI value, the higher the DNA fingerprinting power and the higher the probability of getting unique DNA fingerprint profiles (*Tan et al., 2015*). The obtained total PI $= 2.3 \times 10^{-3}$ in this study is considered low (*Waits, Taberlet & Luikart, 2001*), and hence the markers can be thought as effective for DNA fingerprinting. SSR markers used in Chinese tea cultivars showed a low total PI value of $4.8 \times 10^{-33}$ derived from 312 alleles at 30 loci analyzed on 128 samples (*Tan et al., 2015*), and SSR markers used in Tunisian almond (*Prunus dulcis*) showed a total PI value of $4 \times 10^{-13}$ derived from 159 alleles at 10 loci that were on 82 samples (*Gouta et al., 2010*).

Several factors can influence the ability to construct unique DNA fingerprint profiles, including the number of polymorphic markers and sample size used. Depending on the level of polymorphism of the markers used, the larger the sample size, the more the markers

**Table 7   Summary of analysis of clonal status of nine durian types.**

| Durian type | Sampling locations[a] | Locus | | | | | | |
|---|---|---|---|---|---|---|---|---|
| | | DZ01 | DZ02 | DZ03 | DZ04 | DZ05 | DZ07 | DZ08 |
| D2 | PM, LP, BE, BEA | Same | Different | Same | Same | Same | Same | Same |
| D7 | LP, 5L, BE, BEA | Different | Same | Different | Same | Same | Same | Same |
| D8 | LP, 5L | Same | Same | Same | Same | Same | Same | Same |
| D24 | PM, LP, 5L, BE, BEA | Same | Same | Same | Same | Same | Same | Same |
| D99 | PM, LP, 5L | Different | Different | Same | Different | Same | Same | Same |
| D159 | LP, BE | Different | Same | Different | Different | Same | Same | Different |
| D168 | PM, LP, 5L | Same | Same | Same | Same | Same | Same | Same |
| D188 | LP, BE | Different | Different | Different | Different | Same | Same | Different |
| D197 | PM, LP | Same | Same | Same | Different | Same | Same | Same |

**Notes.**

[a]PM, Putra Mart; LP, Ladang Puchong; BE, Bukit Ekspo; BEA, Bukit Ekspo Plot A; 5L, Ladang 5.

needed. In this study, 21 out of 27 durian types were successfully fingerprinted with only five SSR loci, demonstrating the effectiveness of these SSR markers for fingerprinting of durian types. Still, comprehensive studies that include exhaustive sampling of all registered durian types for a country or a region and more markers are necessary for evaluation of the feasibility of using DNA fingerprinting in the management of registered durian types.

Like many other plants, durian can be either sexually (i.e., via seed) or asexually propagated. Nevertheless, asexual propagation techniques such as cleft grafting, approach grafting, and budding are more commonly practiced to propagate durians so that the quality and consistency of the fruit are preserved (*Abidin, 1991*; *Wiryanta, 2007*). Six durian types (i.e., D2, D99, D197, D159, D188, and D7) showed inconsistent DNA fingerprints across orchards, proving that they are not clones, as clones should be identical in their genetic makeup. It is possible that individuals with different genotypes still produced similar fruits, causing them to be categorized as the same type. Such findings not only showed the utility and importance of DNA fingerprinting in the identification of durian types, but also pose questions on the existing system for the management of durian genetic resource in the region.

## Implications for the management of durian genetic resource

DNA fingerprinting using SSR markers is very useful in assisting the determination of a newly registered variety for Plant Variety Protection (PVP) application (*Silva et al., 2012*), and acting as a tool to complement the assessment of morphological characters (*Treuren et al., 2010*). Apart from using it in new plant variety registration, it can be used to evaluate currently registered plant varieties to investigate if there are clones among registered types. This is particularly important in PVP, as the owner of a new plant variety has the exclusive sale of the plant and exploitation of the plant by the others is illegal. Such DNA fingerprinting method has been used in fingerprinting some important economic crops such as olive cultivars in Turkey (*Ercisli, Ipek & Barut, 2011*), apple cultivars in the Netherlands (*Treuren et al., 2010*), and sugarcanes in Brazil (*Silva et al., 2012*). Therefore, it

is important to determine their identification at a genetic level to ensure that the exported durians are true to a certain type.

The terms "clone" and "variety" are commonly used to refer to the different durian types (e.g., *Abidin, 1991*; *Department of Agriculture Malaysia, 2017*; *Jawahir & Kasiran, 2008*), but each of these terms has a different meaning and should not be used interchangeably. By definition, a "clone" refers to an individual derived from another individual by asexual propagation (*Biosciences for Farming in Africa, 2016*), and so cloned individuals are genetically identical to another. A "variety" means a "plant grouping" that has a set of common characteristics within a species. The term "variety" is not used to refer to a single plant, a trait, or a plant breeding technology (*International Union For The Protection of New Varieties of Plants, 2010*). Therefore, there is a need to reconsider the classification of the durian types we have today, especially by the authority. Whether a registered type should be called a "clone" or a "variety" is not a matter of preference; it affects other aspects related to the adoption of such classification, e.g., the legality revolving the rights to a registered type. If the current situation remains, it is likely that the various durian types are different "varieties" or "cultivars", which are plants with a common set of characteristics, rather than "clones". Then again, this poses a whole new challenge to register, preserve, and validate the authenticity of the various types of durian in the market.

## CONCLUSION

Our results indicated that the SSR marker is a powerful tool to assess the genetic variability in durian. High levels of genetic diversity ($H_E = 0.35$) found in durian in this study provide a foundation for management of genetic resources for the future development of strategies for germplasm sampling and genetic improvement of durian. The results also demonstrated the effectiveness of using SSR markers to genetically fingerprint durian, with 21 out of 27 durian types being successfully fingerprinted using just five markers. The analysis of durian types across orchards has also confirmed that some are not clones, although the samples were claimed to be of the same durian type, challenging the current classification method of durian types in the region.

## ACKNOWLEDGEMENTS

We would like to thank the University Agricultural Park of Universiti Putra Malaysia for allowing us to access the orchards to collect the durian leaf samples.

### Funding

This work was funded by the Universiti Putra Malaysia GP-IPS grant (GP-IPS/2016/9473200). The funders had no role in study design, data collection and analysis, decision to publish, or preparation of the manuscript.

## Grant Disclosures

The following grant information was disclosed by the authors:
Universiti Putra Malaysia GP-IPS: GP-IPS/2016/9473200.

## Competing Interests

The authors declare there are no competing interests.

## Author Contributions

- Ging Yang Siew performed the experiments, analyzed the data, prepared figures and/or tables.
- Wei Lun Ng conceived and designed the experiments, analyzed the data, contributed reagents/materials/analysis tools, prepared figures and/or tables, authored or reviewed drafts of the paper.
- Sheau Wei Tan and Swee Keong Yeap conceived and designed the experiments, contributed reagents/materials/analysis tools, authored or reviewed drafts of the paper.
- Noorjahan Banu Alitheen contributed reagents/materials/analysis tools, authored or reviewed drafts of the paper.
- Soon Guan Tan contributed reagents/materials/analysis tools, authored or reviewed drafts of the paper, arrange for sampling material.

## Data Availability

The SSR genotyping data has been provided as a Supplemental File.

## Supplemental Information

Supplemental information for this article can be found online at http://dx.doi.org/10.7717/peerj.4266#supplemental-information.

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
