# Peer review of "Genetic variation and DNA fingerprinting of durian types in Malaysia using simple sequence repeat (SSR) markers"

_PeerJ, doi:10.7717/peerj.4266_

## Round 0.1 · original submission · Minor Revisions

P5, Lines 12-17, define how to classify different types, based on phenotype?
P5, L20: define CTAB
P6, L5: how to you select the sequence? Are these randomly distributed in the genome? Do you know what chromosomes are they located?
P6. L8, what is the ‘e’ before PCR?
P6, L9: Are you sure you only use 10 ng DNA for 20ul reaction? It is low. You may need to describe how did you quantify DNA.
P6,L23: where is the DNA ladder from? List the company name
P9, L8-9, For these samples without unique fingerprint, do they belong to one of 21 types? If so, these with same genotypes have the same phenotype? Or phenotypically classified in the same type?

P9, L12-16: “Six types (i.e. D2, D99, D197, D159, D188, and D7) were found to contain samples with different fingerprint profiles, with alleles differing at one or more loci. Only three types (i.e. D24, D168, and D8) were found to have consistent fingerprint profiles across orchards.” The sentence is not clearly described. How to determine the 9 types? phenotypically? The six types show different alleles among orchards? The three types shared the same fingerprint among types or among orchards?

Based on the data from this study, can you come up with marker profiles for different types that can be used for classification of these types? The markers were also evaluated for their utility to fingerprint the different durian types.

Reviewer 1 ·

Basic reporting

1. There is no problem with professional English language writing in this manuscript.
2. The abstract clearly summarized the results and addressed the importance of this study. Please add one or two sentences for the content in discussion, which is also a crucial part of your manuscript.
3. In the introduction, the author described basic information about durian, the need and the current progress for genetic characterization for durian. The author also talked very briefly about the objective and significance of this study. The literatures are well referenced and relevant.
Detailed comments on introduction for improvement:
There are few information the author mentioned in latter part of this manuscript that could be moved into introduction. Also, I would suggest the author to add a few information for clarification.
1) Move line 257-260 to introduction paragraph one.
2) Introduce durian’s reproduction biology characteristics, such inbreeding or outbreeding, clones or population.
3) Introduce durian’s polyploidy-level and no. of chromosomes or even genome size (if available).
4) Introduce if there is any next generation sequencing project done for durian (There is a draft genome just published one month ago). If no, the author could claim that the current study is the most advanced study in durian genetic variation study to date. If yes, the author could say that the next step is to utilize the available next generation sequencing data to develop more genetic markers. Either way, the introduction would be more informative.
5) It would be better if the author can summarize what aspects will be discussed in the discussion at the end of introduction.
4. Tables are relevant, well made and footnoted.
5. Raw data are supplied.

Experimental design

The current research is original and within scope of the journal. The research questions are well defined, relevant and meaningful.
Detailed comments for improvement:
1. The sample size (27 durian type) and number of makers (7 SSR markers) are small. It is acceptable in this case as the genetic resources for durian is limited. I would suggest the author to point out this situation either in introduction or discussion. This will help your research to stand out.
2. The author should describe if the varieties you characterized are the same or different from Sales (2015). This will define the uniqueness of the study.
3. The author should address the representativeness/importance of the varieties you used in this study. How much of the area in Malaysia are using these varieties.

Validity of the findings

The fact that only 7 SSR markers can distinguish 21 durian type is significant. The author listed detailed information of the DNA fingerprinting which is very helpful for understanding the context. Also, this validated the conclusion.

The discussion covered a various topics that related to this study, which makes this study more meaningful. The discussion also referred examples in other crops, situations in other countries and compared with recent studies, which makes the context well supported.

Detailed comments for improvement:
The author didn’t describe the morphological difference of these 27 durian types. Do they look similar or different? Does the SSR markers help to distinguish (or corresponding with) different phenotype?

Additional comments

Generally, this study is valid and significant. The manuscript is well written. A few improvements are needed as suggested above.

·

Basic reporting

Literature references:

Previous studies in this genus using different genotype markers are properly cited and compared. However, another paper (Santoso et al. Development of Simple-Sequence Repeats Markers from Durian (Durio zibethinus Murr. cultv. Matahari) Genomic Library) that also used SSR as the genetic marker in Durian just came out. Would be nice to add it to the background section (Line 77-80) .

Grammar:
This is minor - a subject-verb disagreement that I’ve spotted:

Line 58: While the ease of application of these markers make them.. (make -> makes)

Experimental design

No comment

Validity of the findings

No comment

Additional comments

In this study, Siew et al. have investigated seven SSR regions in different types of Durian. With the genetic data in hand, they were able to distinguish 21 different SSR combinations from 27 samples, and genetic diversity was observed within the same Durian species. This manuscript is well-prepared, and the results are interesting regarding to plant identification as well as other implications.

The authors do a good job presenting the rationales of using SSR as a DNA barcoding tool in Durian. The question is well defined, and the methods are standard and acceptably presented. Molecular cloning procedures, details and primers are provided for others to replicate the same experiments.

Minor point:

SSR variations were observed in the same type of Durian that were obtained from different orchards (Line 167-182), which is potentially interesting. Genetic distance plot, such as dendrogram or PCA analysis, will help to clarify the genetic relationship between types and geographic locations.

---

## Round 0.2 · Minor Revisions

My suggested changes and comments are marked in the manuscript.

---

## Round 0.3 · accepted · Accept

Dear Wei Lun and Noorjahan Banu,

Thank you for your submission to PeerJ.

I am writing to inform you that your manuscript - Genetic variation and DNA fingerprinting of durian types in Malaysia using simple sequence repeat (SSR) markers - has been Accepted for publication. Congratulations!

We are now checking your article for production and you will soon receive a list of Production tasks.

Congratulations again, and thank you for your submission.

With kind regards,
Guihua Bai
Academic Editor, PeerJ